# Situational Analysis and Knowledge, Attitudes, and Practices of Antimicrobial Use and Resistance among Broiler Poultry Farmers in Nepal

**DOI:** 10.3390/ani13193135

**Published:** 2023-10-07

**Authors:** Manoj Kumar Shahi, Tulsi Ram Gompo, Sumit Sharma, Bishal Pokhrel, Srijana Manandhar, Saharuetai Jeamsripong

**Affiliations:** 1Nepal Veterinary Council, Tripureshwor, Kathmandu 4600, Nepal; avisanu2006@gmail.com (M.K.S.); shrijanam@yahoo.com (S.M.); 2Central Veterinary Laboratory, Tripureshwar, Kathmandu 44600, Nepal; 3Department of Veterinary Public Health, Faculty of Veterinary Science, Chulalongkorn University, Henri-Dunant Rd., Pathumwan, Bangkok 10330, Thailand

**Keywords:** antimicrobial resistance, antimicrobial use, broiler poultry farmer, KAP, Nepal

## Abstract

**Simple Summary:**

The objective of this study was to evaluate the knowledge, attitudes, and practices (KAP) of broiler poultry farmers (BPF) in Nepal concerning the use of antimicrobials (AMU) and the issue of antimicrobial resistance (AMR). A total of 500 farmers from all seven provinces of Nepal, spanning 40 districts, participated in this study. The collected data, encompassing AMU, demographic distribution, and KAP regarding AMU and AMR among BPF, underwent descriptive statistic and logistic regression analysis. This study’s results revealed that BPF possessed limited understanding and engagement in practices related to AMU and AMR, whereas they exhibited positive attitudes. This highlights the importance of implementing effective regulatory measures, improving veterinary services, and providing training and awareness programs to address the issue of AMR in the poultry farming sector.

**Abstract:**

The purpose of this study was twofold: first, to conduct a situational analysis, and second, to assess the knowledge, attitudes, and practices (KAP) of broiler poultry farmers (BPF) regarding antimicrobial use (AMU) and antimicrobial resistance (AMR). Data were collected from 500 BPF across 40 districts, representing 88.1% of Nepal’s broiler poultry population. Among these farmers, 81.0% were male, 59.6% had up to 4 years of experience in poultry farming, and 50.8% had completed at least a high school education. The most used antimicrobials on the farms were doxycycline (23.5%), neomycin (17.1%), and colistin sulfate (9.6%) out of 27 reported antimicrobials. While the BPF exhibited limited knowledge (62.6%) and practice (55.5%) related to AMU and AMR, their overall attitude toward these issues was positive (91.6%). This study also identified significant factors influencing farmers’ attitudes toward AMU and AMR. Farmers aged 31–40 showed a stronger inclination compared to other age groups (OR = 4.2, *p* = 0.02), and those using antimicrobials for preventive purposes had a more favorable attitude compared to those using them for other purposes (OR = 5.9, *p* = 0.02). In light of these findings, this study recommends the implementation of effective regulatory measures for drug usage, along with awareness programs addressing AMU and AMR to address the issue of AMR in poultry production.

## 1. Introduction

The global concern regarding antimicrobial resistance (AMR) has gained significant importance, particularly within the context of the One Health approach. The improper use of antimicrobials in both human and veterinary medicine is leading to the emergence of resistant bacteria [1]. The emergence of AMR in animal production has consequences for human health through the food production chain [2]. On a global scale, AMR is responsible for 700,000 human deaths annually. If immediate action is not taken, it is estimated that by 2050, the economic cost of AMR infections could reach USD 100 trillion, potentially resulting in approximately 10 million human deaths [3]. Low- and middle-income countries are grappling with more significant challenges related to AMR compared to developed countries [4]. Consequently, raising awareness about antimicrobial usage (AMU) among various stakeholders involved in livestock production can be a potent strategy for managing and preventing AMR.

Antimicrobials play a crucial role in managing and controlling bacterial infections and promoting growth in food-producing animals, such as poultry [5]. However, the use of antimicrobials as growth promoters in animal feed has been prohibited in several countries like Sweden and the United States [6,7]. Furthermore, Nepal banned the use of a growth promoter in 2017 [8]. The use of antimicrobials in poultry has led to favorable outcomes, including a decrease in disease occurrence, reduced morbidity and mortality rates, improved animal health, increased productivity, and higher economic returns [9]. Nonetheless, employing antimicrobial agents may result in various adverse consequences for human, animal, and environmental health. This impact encompasses the emergence of AMR, heightened health hazards for both animals and humans, diminished choices in antimicrobial treatments, environmental contamination, economic repercussions, and potential global health crises.

In Nepal, commonly used antimicrobials for poultry production encompassed amikacin, gentamicin, streptomycin, cefotaxime, ceftazidime, ciprofloxacin, ampicillin, colistin, sulfamethoxazole, and tetracycline [8]. However, improper antimicrobial usage can create an environment conducive to the development of resistant bacteria, which can transfer their resistant determinants within and among different bacterial species [10,11]. Previous studies have reported the characterization of both phenotypic and genotypic resistance profiles in strains isolated from poultry, including *Escherichia coli* and *Enterococcus faecalis* [12,13]. The efficacy of available antimicrobials is gradually diminishing due to the global dissemination of multidrug resistance (MDR) mechanisms in both commensal and pathogenic bacteria [5].

Poultry meat and eggs play a significant role as a primary source of animal protein in Nepal, contributing to the country’s economic growth. In 2021, the Nepalese government achieved self-sufficiency in poultry meat and egg production. While Nepal’s Ministry of Health and Population approved the National Antibiotic Containment Action Plan in 2016 and the National Antibiotic Treatment Guideline in 2014, both aimed at human health [14], it is worth noting that Nepal’s Drug Act of 1978 lacks provisions for regulating veterinary drugs. Given this situation, the responsible use of antimicrobials in animals in Nepal is of utmost importance.

The knowledge, attitude, and practice (KAP) study is a widely employed quantitative research method that helps uncover misconceptions or misunderstandings that may pose challenges in altering human behavior and potentially act as barriers to change [15]. The process of modifying human behavior involves three consecutive stages: acquiring knowledge, shaping attitudes, and developing behaviors [16]. A KAP survey involves investigating the behaviors of a representative target group to gather information about their understanding, beliefs, and actions concerning a specific topic [15]. The lack of awareness about AMU and AMR among various stakeholders involved in poultry production can exacerbate the AMR issue, potentially posing risks to public health. The rational use of antimicrobials is closely linked to the KAP of farmers, who are the end-users of antimicrobials. This connection, guiding the farmer, is a crucial factor in the success of efforts to prevent and control AMR.

In the realm of livestock production, the main factors contributing to the development of AMR in animals include farmers’ tendency to self-prescribe medications, unauthorized antimicrobial use (AMU), and the absence of regulations governing AMU. To effectively mitigate AMR, several key measures should be taken, including promoting the responsible use of antimicrobials, monitoring the prevalence of AMR, raising awareness about AMR, encouraging efficient antimicrobial practices, enacting relevant legislation, and establishing regulatory mechanisms to curb AMR. While more data are yet to be reported regarding the economic and livestock health impacts of AMR in developing countries, it is evident that veterinarians, as integral components of the AMR solution, should undergo comprehensive training and operate under the supervision of veterinary regulatory bodies [17,18]. Understanding the practices of AMU among farmers is crucial because farmers’ KAP significantly shapes the patterns of AMU in livestock farming [19]. Therefore, the aim of this study is to assess the state of AMU in broiler poultry farms and evaluate the KAP pertaining to AMU and AMR among farmers involved in broiler poultry production in Nepal. The results of this study have the potential to offer valuable insights to policymakers in their efforts to address AMR both at the national and global levels.

## 2. Materials and Methods

### 2.1. Questionnaire

The questionnaires were structured to encompass various aspects, including demographic details, an assessment of AMU conditions, and an evaluation of KAP associated with AMU and AMR among farmers. The majority of the questions were in a multiple-choice format. Demographic information covered aspects such as age, gender, educational level, work experience, province, and ecozone. Questions related to AMU delved into various aspects of poultry management, including the type, frequency, source, and storage of antimicrobials, as well as farm management regarding flock size, bird health, farm biosecurity, mortality rates, clinical signs, and disease occurrence. Knowledge-related questions focused on topics such as AMU, withdrawal periods for antimicrobials, AMR transmission, and government policies regarding AMU. In terms of perceptions about AMU and AMR, the attitude section explored aspects such as the safety of AMU and the use of nonprescribed antimicrobials. Finally, the practice-related questions sought information on completing full courses of AMU, skipping doses, the frequency of AMU, and the verification of antimicrobial expiration dates.

### 2.2. Validation of the Questionnaire

Before commencing the actual survey, a pre-test of the questionnaires was conducted, involving the random selection of five farmers. Following this, the questionnaire underwent validation by being submitted to three experts in the field. These experts assessed the questionnaire items for their congruence with the study objectives using the item-objective congruence (IOC) index. For each section of the questionnaires, the IOC point was assigned a rating. To assess the consistency and congruence of all elements, experts were presented with a three-scale rating, and they were required to provide a single rating from these options. The overall IOC value of this set of questionnaires was determined to be 0.9.

### 2.3. Study Area

This study involved 500 broiler poultry farmers (BPFs) selected from 40 districts across Nepal’s seven provinces, namely, Sudurpaschim, Karnali, Lumbini, Gandaki, Bagmati, Madhesh, and Koshi. The selection criteria were based on the areas with the highest broiler poultry population density in the country. At least three districts from each province and ecozone of Nepal were selected for this study. In total, the study area covered 88.1% of Nepal’s broiler population.

Specifically, the selected districts, which were all included in the farmer questionnaire survey, included the following: Sudurpaschim province (Bajura, Kailali, and Kanchanpur); Karnali province (Surkhet, Dailekh, and Jajarkot); Lumbini province (Banke, Bardia, Dang, Gulmi, Palpa, Pyuthan, and Rupandehi); Gandaki province (Baglung, Gorkha, Kaski, Nawalparasi East, Syangjha, and Tanahun); Bagmati province (Bhaktapur, Chitwan, Dhading, Kathmandu, Kavreplanchok, Lalitpur, Makawanpur, Nuwakot, Sindhuli, and Sindhupalchok); Madhesh province (Bara, Mahottari, Rautahat, Saptari, and Siraha); and Koshi province (Ilam, Jhapa, Morang, Okhaldunga, Sunsari, and Udayapur) (Figure 1).

### 2.4. Study Population and Sample Size Determination

In Nepal, there were 52.7 million broiler poultry distributed across 20,483 farms in the year 2017. Among the selected 40 districts, there were a total of 46.4 million broiler poultry and 17,601 farms [20]. In this context, the term “farmers” refers to individuals involved in poultry farming, including owners, staff, technicians, or veterinarians responsible for farm management and medication. The number of survey participants per district was determined based on the proportion of the broiler population, with one respondent per farm.

To calculate the total number of farmers/respondents required for the questionnaire survey, a single proportion calculation was performed using a 95% confidence interval and a desired precision of 5%, assuming a 50% estimated response rate. The minimum sample size for respondents was determined to be at least 392 farmers.

### 2.5. Questionnaire Survey and Data Collection

Between May 2022 and May 2023, a cross-sectional study involving broiler poultry farmers was conducted. The data collection process involved face-to-face interviews with the BPF using a semi-structured questionnaire. These interviews were carried out by a team of 46 trained veterinarians. All participating veterinarians involved in the farmer survey received orientation, with some attending in-person sessions across various locations within the seven provinces, while others received their orientation through online resources, video tutorials, or phone consultations. Initially, the questionnaires were prepared in English and were subsequently translated into Nepali for data collection purposes. All the data were recorded using EpiCollect5 (http://five.epicollect.net, accessed on 2 October 2023). Prior to participating, each farmer received an explanation of the survey’s objectives and purpose. The inclusion in this study was restricted to those farmers who provided their consent. To validate the information regarding AMU supplied by the farmers, records, antimicrobial photos, and antimicrobial labels or packaging materials were used.

### 2.6. Statistical Analysis

The data collected through the questionnaire survey underwent several steps for processing and validation in Microsoft Excel 365. Responses to the questionnaire were assigned numerical codes from 1 to 5 for “Completely agree”, “Agree”, “Disagree”, “Completely disagree”, and “Do not know”. These codes were used to calculate the overall KAP score. To categorize KAP scores as either “Good” or “Not good”, the median value was computed for each respondent based on their responses. If a respondent’s overall median value was below or equal to the calculated median, their KAP was considered “Good”, while a value above the median indicated a “Not good” KAP level.

Descriptive analysis was carried out to provide an overview of the AMU situation and KAP among farmers. Logistic regression analyses were used to identify associations between variables related to KAP and the outcome variables of AMU and AMR. Independent variables with a *p*-value less than 0.1 in the univariate analysis were selected for inclusion in the multivariate analysis. The results of univariate logistic regression analysis were considered statistically significant if the *p*-value of the variables in the final multivariate analysis was *p* < 0.05. All statistical analyses were performed using STATA/SE 14 (StataCorp, College Station, TX, USA), and QGIS 3.4 (https://www.qgis.org/en/site/, accessed on 2 October 2023) was used to depict the spatial distribution of sampling locations on a map.

## 3. Results

### 3.1. Demographic Characteristics of BPF

Out of the 500 participating BPF, their distribution across different regions of Nepal was as follows: Terai region (53.8%), Hill region (42.2%), and Mountain region (4.0%) of Nepal (Table 1). The highest number of participants originated from the Bagmati province (31.6%), followed by Lumbini (17.2%), Koshi (17.0%), and Gandaki (12.6%), while the lowest number of participants came from the Karnali provinces (5.4%). The majority of the BPF were male (81.0%) and had an average age of 36.7 ± 9.0 (31–40) years (41.4%). In terms of experience, most farmers had 0–4 years of experience (59.6%), possessed a high school education (50.8%), and had completed primary education (26.8%).

Concerning farm size, 57.8% of the farms were categorized as small (<1500 birds), while 29.4% and 12.8% were classified as medium (1501–5000 birds) and large (>5000 birds), respectively (Table 1). The average flock size in these poultry farms was 3155 ± 10,967.0 (250–178,000) birds. Nearly all farmers (98.6%) reported using commercial feed for their birds. Regarding water sources, most respondents relied on groundwater (47.4%), followed by municipal supply (41.2%), and a smaller proportion used deep wells (10.8%). Furthermore, the average mortality rate among broiler poultry flocks was 9.3 ± 15.5%, with values ranging from 0% to 100%. On average, the income per 100 birds per batch in broiler poultry farming amounted to USD 43.8 ± 22.6 (0–105.7) and USD 1617.2 ± 279.0 (0–102,459.0), respectively, while the average cost of medicine per 100 birds was reported as USD 6.9 ± 7.5 (0–49.8).

### 3.2. Antimicrobial Use in BPF

In Nepal, nearly all BPF had easy access to antimicrobials within the study area, primarily attributed to the availability of over-the-counter sales of veterinary drugs. The poultry production sector employed a total of 27 different types of antimicrobials belonging to 13 distinct classes. Among these, the most commonly used class of antimicrobials was tetracyclines (28.0%), followed by aminoglycosides (21.5%), quinolones (13.0%), and polymyxins (9.6%) (Figure 2). Concerning antimicrobial combinations, the neomycin–doxycycline combination (22.4%) was the most frequently used, followed by colistin sulfate–amoxicillin (4.8%). Colistin sulfate was often used in combination with amoxicillin, gentamicin, tylosin, doxycycline, and tetracycline, while doxycycline was frequently combined with gentamicin, neomycin, and tylosin.

On average, during each cycle of broiler production, antimicrobials were employed approximately 1.6 (0–4) times across 781 cases of 500 responses. Approximately 9.2% of the farmers reported never using antimicrobials throughout the production cycle, while the remaining 90.8% of farmers used antimicrobials at least once. Among the 908 multiple responses, the most used antimicrobials were doxycycline (23.5%), neomycin (17.1%), and colistin sulfate (9.6%) (Figure 3).

Regarding the purpose for which antimicrobials were used in animal feed, approximately 72.7% of farmers used them for treating poultry diseases, followed by prevention and control (18.2%) and growth promotion (3.1%). Approximately half (50.1%) of the farmers obtained antimicrobials directly from drug sellers. In contrast, only 33.4% and 10.3% of farmers purchased antimicrobials with prescriptions from veterinarians and para-veterinarians, respectively. Based on frequently observed clinical signs, digestive system-related signs (39.6%) were the most commonly reported, followed by respiratory system-related signs (37.7%). The use of antimicrobials in broiler production predominantly occurred during the first 3 days (16.6%) and slightly during days 4–10 (12.3%). Antimicrobial usage peaked between days 10 and 20 (25.4%), gradually decreasing to 21.9% and 23.8% on days 21–30 and >30 days, respectively. When birds exhibited illness or signs of disease, 53.2% of farmers sought advice from veterinarians, while 21.6% consulted drug sellers, 20.0% consulted para-veterinarians, and 4.4% engaged in self-treatment (Table 2).

### 3.3. KAP of BPF on AMU and AMR

#### 3.3.1. Knowledge of BPF on AMU and AMR

A significant portion of farmers (62.6%) demonstrated good knowledge concerning AMU and AMR. A substantial majority (74.8%) either completely agreed or agreed with the idea that different antimicrobials have varying curative effects on various poultry diseases. About three-quarters of BPF (72.0%) were aware that antimicrobials used in poultry production could potentially be transmitted to humans through the consumption of poultry meat and eggs. Furthermore, 71.6% of farmers acknowledged that the presence of antimicrobial residues in poultry meat could pose hazards to human health. Similarly, 70.6% of respondents recognized that poultry should be sold after adhering to the proper antimicrobial withdrawal period to prevent the occurrence of antimicrobial residue in the meat. The results indicated that the majority (50.4%) of BPF were aware that the use of antimicrobials in animal feed is inappropriate. However, only 41.2% of farmers were aware of the government’s policies and plans related to AMU and AMR. Interestingly, a small proportion of farmers (16.0%) did not administer treatment to the entire flock when only one or a few birds were affected (Figure 4).

#### 3.3.2. Attitude of BPF toward AMU and AMR

This study revealed that most farmers (91.6%) held a positive attitude toward AMR and AMU. Approximately three-quarters of BPF (73.4%) believed that expired antimicrobials should not be administered to birds. Most respondents (72.2%) had a positive view of vaccination as a potential strategy to reduce the reliance on antimicrobials in poultry farming. Furthermore, a significant proportion of respondents (68.0%) advocated for the use of antimicrobials solely for preventing severe illnesses. Similarly, 60.0% of farmers acknowledged that improper doses of antimicrobials could contribute to AMR. In contrast, 54.6% of respondents believed that antimicrobials were unnecessary for treating fever or cold in humans. Additionally, nearly half of them expressed that antimicrobials were not needed for birds during seasonal changes (46.6%), and 41.2% believed that antimicrobials were unnecessary for treating any animal disease. Lastly, 24.2% of the respondents expressed concerns about the adverse effects of AMU on animal health (Figure 5).

#### 3.3.3. Practice of BPF on AMU and AMR

This study revealed that 55.5% of farmers exhibited commendable practices when it came to AMU and AMR. Most farmers (93.2%) adhered to the complete course of antimicrobial treatment as prescribed by veterinarians. Additionally, almost all farmers (88.0%) routinely checked the expiration dates of antimicrobials before using them. Two-thirds of farmers (66.8%) did not practice the habit of skipping one or two doses of their prescribed antimicrobial courses. Approximately 43.0% of respondents did not resort to increasing the dosage and frequency of prescribed antimicrobials when clinical symptoms failed to disappear or subside, and a similar proportion of farmers (42.0%) did not continue AMU when symptoms had disappeared. However, 37.2% of farmers indicated a preference for not using antimicrobials as a preventive measure against disease (Figure 6).

### 3.4. Factors Affecting KAP on AMU and AMR of BPF

In the univariate analysis, several factors were associated with knowledge of AMU and AMR, including province, age group, purpose of antimicrobials, control of antimicrobial sales, belief in the role of vaccination in reducing AMU, and the source of advice for antimicrobials (Table 3). Farmers residing in the Bagmati province demonstrated significantly higher knowledge levels regarding AMU and AMR when compared to farmers in the Lumbini (0.2, *p* = 0.008), Koshi (0.2, *p* = 0.009), and Sudurpaschim provinces (0.2, *p* = 0.04). Moreover, farmers in the Koshi province exhibited 2.9 times better AMU practices than farmers in the Bagmati province (*p* = 0.006). Farmers within the 31–40 age group displayed a notably more positive attitude, which was 4.2 times better than their counterparts in the 10–30 age group. Furthermore, farmers who used antimicrobials for treatment purposes exhibited practices that were 2.1 times better than those using antimicrobials for all purposes (*p* = 0.01).

Farmers who agreed with the statement “control of antimicrobial sales contributes to AMR” demonstrated significantly higher knowledge levels, which were 14.5 times greater than those who disagreed (*p* = 0.00001). Similarly, farmers who believed in the statement “vaccination can reduce AMU” displayed better knowledge (odds ratio (OR) 3.5, *p* = 0.002) and attitude (OR 15.0, *p* = 0.0001) compared to farmers who did not share this belief. Additionally, farmers who sought advice on AMU from veterinarians exhibited practices that were 21.0 times better than those seeking advice from drug sellers (*p* = 0.0001).

The multiple logistic regression analysis identified several factors associated with AMU and KAP among BPF in Nepal (Table 4). Variables including province, gender, age, purpose of AMU, control of antimicrobial sales, vaccination, and source of advice were found to be associated with KAP related to AMU and AMR. For example, farmers residing in the Lumbini (OR = 0.2, *p* = 0.008), Koshi (OR = 0.2, *p* = 0.009), and Sudurpaschim (OR = 0.2, *p* = 0.004) provinces showed significantly lower levels of knowledge concerning AMU and AMR compared to farmers in the Bagmati province. Furthermore, farmers in the Bagmati province had a fivefold (1/0.2) higher level of awareness regarding AMU and AMR in comparison to those in the Lumbini province (*p* = 0.03). Farmers in the Koshi province demonstrated significantly better practices compared to those in the Bagmati province (OR = 2.9, *p* = 0.006). In terms of age, farmers in the 31–40 age group displayed attitudes toward AMU and AMR that were 4.2 times more favorable compared to those in the 18–30 age group (*p* = 0.02). 

In terms of the purpose of AMU in poultry production, farmers who were unaware of the specific purpose of AMU demonstrated 16.4 times more knowledge about AMU and AMR than farmers who used antimicrobials for all purposes (*p* = 0.02). Additionally, farmers who used antimicrobials for prevention and control purposes exhibited a 5.9 times more favorable attitude toward AMU and AMR compared to farmers who used antimicrobials for all purposes in poultry (*p* = 0.02). Interestingly, farmers who used antimicrobials to treat their poultry exhibited better practices in AMU than those who used antimicrobials for all purposes (*p* = 0.01).

Regarding the perception of antimicrobial control and AMR occurrence, farmers who believed that the lack of antimicrobial sales control could contribute to AMR had 14.5 times greater knowledge and 2.6 times more favorable attitudes about AMU and AMR than their counterparts (*p* < 0.0001). Similarly, those who had faith that vaccines could reduce AMU demonstrated significantly better knowledge and attitudes toward AMU and AMR than those who did not believe in the reduction of AMU through vaccination, with *p* = 0.002 and *p* < 0.0001, respectively. Regarding the sources of advice for poultry health, farmers who consulted with veterinarians when their poultry were sick exhibited better practices regarding AMU than those who consulted with drug sellers only (OR = 21.0, *p* < 0.0001). Farmers who practiced self-treatment displayed 8.7 times better practices in AMU than those who consulted with drug sellers (*p* = 0.001).

### 3.5. Association among the KAP on AMU and AMR of BPF

The results indicate a significant positive correlation between the level of knowledge regarding AMU and AMR among BPF and their corresponding attitudes and practices (Table 5). Specifically, BPF who possessed a strong understanding of AMU and AMR demonstrated attitudes that were 19.4 times more favorable (*p* < 0.0001). Furthermore, there was a direct association between the depth of knowledge of farmers regarding AMU and AMR and their actual practices (OR = 1.7, *p* = 0.004). However, the association between farmers’ attitudes and practices was not found to be significant. This suggests that having a positive attitude toward AMU and AMR may not necessarily translate into corresponding practices.

## 4. Discussion

This study provided significant insights into the KAP of poultry farmers regarding AMU and AMR in Nepal’s poultry production. The findings underscored the importance of implementing policies that promote the responsible use of antimicrobials in poultry farming and raising awareness among key stakeholders within the framework of the One Health concept to effectively address AMR issues. Additionally, the research highlighted the necessity of fostering sustainable practices and attitudes related to AMU and AMR to address ongoing challenges posed by AMR and ensure the long-term health and sustainability of poultry production.

Regarding specific observations, this study noted that the average mortality rate among broiler poultry flocks was 9.3 ± 15.5%, slightly lower than in previous studies [21]. Additionally, this study found that the average income per 100 birds was USD 43.8 ± 22.6, with a range from USD 0 to 105.7. This relatively low income can be attributed to the significant expenses associated with imported feed and its raw materials for the birds, which was exacerbated by importation barriers imposed by the Nepali government in 2021 due to economic challenges. In the context of poultry production in Nepal, this study identified a total of 27 different types of antimicrobials, consistent with previous studies [22,23].

This study identified the prevalent classes and types of antimicrobials utilized in broiler poultry farms in Nepal, with tetracyclines being the most commonly used, followed by aminoglycosides, quinolones, polymyxin, penicillin, macrolides, and sulfonamides. These findings align with similar observations made in other low- and middle-income countries (LMICs). For example, in the Kathmandu and Chitwan districts, a prior study reported that the most frequently used antimicrobials were doxycycline (25.9%), tylosin (21.5%), colistin (18.8%), ciprofloxacin (13.4%), and neomycin (12.5%) [24]. In Pakistan, during the summer season, the common antimicrobials in use are neomycin, doxycycline, and tilmicosin, whereas in the winter, doxycycline, neomycin, and ampicillin are frequently used [25]. Notably, more than 60% of AMU in broiler chicken in the same study was associated with critically important antimicrobial classes for human medicine [25]. A 5-year surveillance of AMU in Pakistan revealed that colistin, tylosin, doxycycline, and enrofloxacin are frequently used on broiler chicken farms [26]. In Nepal, the most consumed antimicrobials include tetracycline, enrofloxacin, neomycin–doxycycline, levofloxacin, colistin, and tylosin [22,23], with colistin, doxycycline, and neomycin being commonly used on broiler farms [27].

Nepal has implemented regulations governing AMU in food animals, encompassing the Drug Act of 1967, the National Drug Policy introduced in 1995, and the Drug Sales and Distribution Code from 2014. Additionally, they have established a National AMR Containment Action Plan in 2016 [28]. Nevertheless, worldwide AMU continues to rise, particularly in low- and middle-income countries (LMICs), with India and China at the forefront of this trend [29,30]. In India, consistent guidelines are lacking, although there was a recommendation in 2007 to discontinue the use of systemic antibiotics in animal feed [31,32]. China has introduced various regulations over the past two decades aimed at controlling the AMU in agriculture [33,34]. In many LMICs, the endorsement of the National Action Plan (NAP) for containing AMR has been observed. However, the oversight of AMU in animals varies significantly depending on the specific circumstances of each country. For example, Pakistan has launched its national strategic framework to contain AMR, focusing on the preservation of the effectiveness of existing antimicrobial drugs and the reduction of expenses and risks associated with their indiscriminate use [35]. Similarly, the Bangladesh government initiated its NAP for AMR containment in 2017–2020 and prohibited AMU in animal feed through the Bangladesh Fish Feed and Animal Feed Act of 2010. Nevertheless, there is currently no specific drug policy or guideline in place for the appropriate AMU to treat animals [36,37].

Several factors influence AMU, including the availability of antimicrobials, prescription patterns by veterinarians, potential pressure from drug sellers to reduce excess drug inventory, lack of awareness, and the relatively low cost or easy accessibility of certain antimicrobials for farmers. Despite the Nepal government’s prohibition of colistin sulfate in food animals since August 2019, this antimicrobial continues to be used in poultry farming. One possible reason for the ongoing use of colistin could be the belief that these antimicrobials are effective against common bacterial infections in poultry, such as respiratory and gastrointestinal infections. An integrated surveillance approach encompassing Pakistan, Bangladesh, and Nigeria regarding colistin resistance revealed that the practice of using colistin as a growth promoter persists in LMICs, with no signs of decline, despite its critical classification by the WHO [38,39]. These findings underscore the limited awareness of colistin and resistance among farmers, highlighting the pressing need for enhanced public awareness campaigns about AMU within diverse cultural and socioeconomic contexts.

According to this study, most individuals engaged in poultry farming had less than 4 years of experience. This trend may be attributed to experienced farmers leaving the poultry sector due to financial setbacks resulting from various poultry disease outbreaks, such as Newcastle disease, Low Pathogenic Avian Influenza infection, etc. Additionally, the COVID-19 pandemic led to an influx of labor into the poultry sector due to reverse labor migration. Notably, farmers in the age group of 31–40 exhibited a more favorable attitude compared to those aged 18–30. This difference could be due to older farmers being more self-reliant and having greater exposure to training and work experience than their younger counterparts. A previous study on KAP in Bangladesh also highlighted the influence of farmers’ age and experience on AMU and AMR in poultry production [21]. Approximately 9.2% of farmers reported not using antimicrobials throughout the entire broiler production cycle, while 90.8% used antimicrobials at least once during the production cycle, indicating the continued prevalence of AMU in broiler production. Our descriptive analysis revealed that farmers in Karnali province maintain smaller poultry flocks and reside in areas with relatively lower incidences of poultry diseases.

This study has highlighted that farmers, despite having a positive attitude toward AMU and AMR, often lack a comprehensive understanding of the consequences associated with these practices in farm settings. The gap between their knowledge and attitude, resulting in improper practices, can be attributed to several factors, including insufficient education regarding infectious disease control and prevention, limited access to veterinarians, inadequate awareness of AMU and AMR, and less stringent regulations governing AMU in animals. While the majority of farmers demonstrated positive awareness toward the withdrawal periods, the transmission of zoonotic diseases, and the appropriate use of antimicrobials for severe diseases, some farmers did not possess the same level of awareness regarding these critical aspects of AMU and AMR. This suggests that there is room for improvement in education and awareness-raising efforts to ensure that all farmers have a comprehensive understanding of these important concepts and practices. Despite their generally positive attitudes, most farmers (91.6%) exhibited limited knowledge and engaged in improper practices related to AMU and AMR. These knowledge and practice gaps may be attributed to various factors, including the limited availability of veterinarians, diagnostic facilities for disease, and training on the prudent use of antimicrobials and AMR. Notably, a significant proportion of farmers (85.0%) expressed support for the prohibition of non-prescribed antimicrobials for sale. This support may stem from negative past experiences with these drugs and concerns about the impact of AMR on both human and animal health. This statement aligns with our findings on knowledge, with three-quarters of farmers demonstrating awareness of national AMR policies and the concept of withdrawal periods. Additionally, this study revealed that 72.2% of BPF consider vaccination as a potential strategy to reduce the use of antimicrobials in poultry farming. This finding underscores the BPF’s clear understanding of the importance of adopting alternative approaches to reduce AMU in animal production.

In this study, it was observed that 50.1% of farmers purchased antimicrobials directly from the drug sellers, which is consistent with findings from previous studies [23,40]. In comparison, only 33.4% and 10.3% of farmers purchased antimicrobials through prescriptions from veterinarians and para-veterinarians, respectively. There may be instances where farmers voluntarily choose to use antimicrobials, either willingly or influenced by recommendations from drug sellers [21,40]. Additionally, limited access to veterinary services could contribute to farmers seeking advice and medication from non-experts. Approximately 53.2% of the farmers consulted veterinarians for treatment, while the remaining farmers relied on drug sellers (21.6%) or para-veterinarians (20.0%) and engaged in self-treatment (4.4%) for their birds. The main reasons for not seeking veterinary services have been discussed in previous studies and may include factors such as the availability of veterinarians, challenges in accessing laboratory services, seeking advice from neighboring farmers, and the ease of obtaining antimicrobials [21,24,41,42].

Based on logistic regression analysis, the BPF in the Bagmati province exhibited comparatively higher knowledge regarding AMU and AMR compared to those in the Lumbini (OR = 0.20, *p* = 0.008), Koshi (OR = 0.20; *p* = 0.009), and Sudurpaschim (OR = 0.20; *p* = 0.04) provinces. Furthermore, the farmers in the Bagmati province demonstrated more positive attitudes than those in the Lumbini province (OR = 0.20; *p* = 0.03) and better practices than those from the Koshi province (OR = 2.9; *p* = 0.006). This variation could be attributed to several factors, including higher levels of education among farmers in the Bagmati province and their residence in areas where poultry farming is more commercialized. Moreover, they have increased exposure to veterinary research centers, academic institutes, and veterinary training centers, all of which are situated within the Bagmati province [43,44]. Furthermore, this study indicated that farmers who used antimicrobials for treatment purposes exhibited better practices in AMU in poultry farming compared to those who used antimicrobials for all purposes (OR = 2.1; *p* = 0.01). These findings underscore the importance of implementing targeted educational and training programs for farmers. These initiatives should prioritize enhancing farmers’ understanding of AMU, particularly regarding the distinction between different types of use and their specific purposes. By promoting the appropriate use of antimicrobials, farmers can contribute to a more responsible utilization of these substances in poultry production.

A significant number of farmers believed that vaccines could contribute to the reduction of AMU, and this belief is significantly and positively associated with both their knowledge (OR = 3.5; *p* = 0.002) and attitude (OR = 15.0; *p* < 0.0001) toward AMU and AMR. This perception may stem from their awareness of government-led vaccination campaigns aimed at combating major poultry diseases like Newcastle disease, infectious bursal disease, and fowlpox [24]. Therefore, farmers may have a high level of confidence in the effectiveness of vaccines, as these campaigns have demonstrated their ability to safeguard farm economies by reducing mortality rates. Farmers who sought advice from veterinarians for poultry treatment (OR = 21.0; *p* < 0.0001) exhibited better AMU practices than those who consulted with drug sellers. This difference may be attributed to the provision of accurate information and the establishment of more effective learning platforms through veterinary consultations, which ultimately result in improved AMU practices.

Therefore, this study underscores the critical importance of raising awareness among farmers about strategies to mitigate AMU and emphasizes the significance of consulting veterinarians for poultry health management. To promote responsible AMU practices and achieve more sustainable and effective poultry farming, it is recommended to implement targeted educational and training programs. These initiatives can play a vital role in addressing concerns related to AMR and ensuring the long-term viability of poultry production.

## 5. Conclusions

This study underscores the critical challenges confronting the poultry production sector, including limited access to veterinary services, the unregulated use of veterinary medications, farmers prescribing antimicrobials directly, and erratic prescriptions by para-veterinarians. These collective factors, coupled with farmers’ lack of knowledge and inappropriate practices, have significantly contributed to the rise of AMR in poultry farming. To effectively combat this issue, a multifaceted strategy is imperative. It should encompass comprehensive education and training programs targeting all relevant stakeholders, the promotion of alternatives to AMU, stringent regulation of veterinary drugs, and the establishment of monitoring and surveillance mechanisms for both AMU and AMR in poultry farming. This proactive approach is essential for mitigating the problem of AMR within the poultry industry and safeguarding the health of both animals and humans.

## Figures and Tables

**Figure 1 animals-13-03135-f001:**
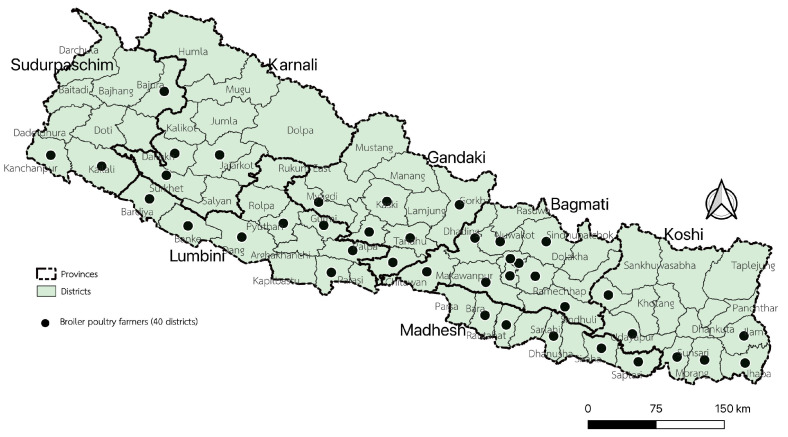
Geographical distribution of study area of Nepal.

**Figure 2 animals-13-03135-f002:**
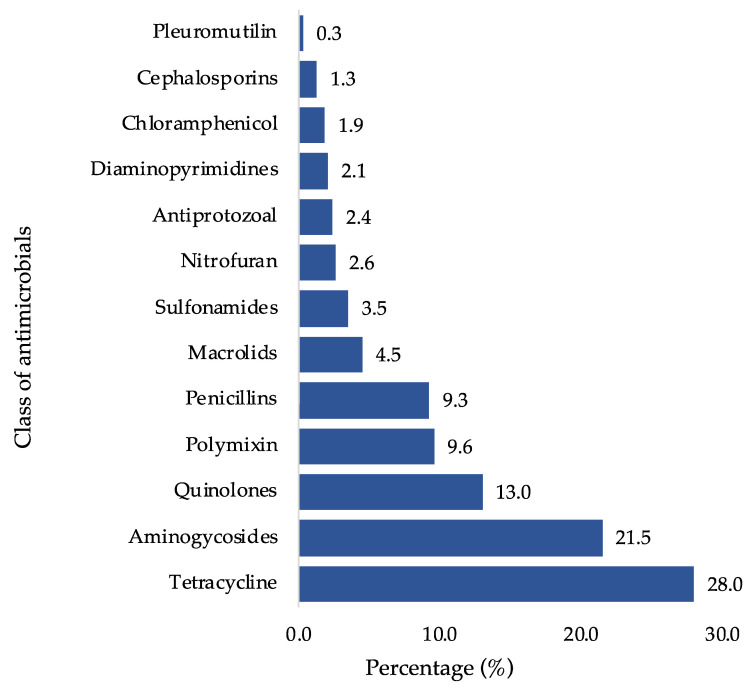
Classes of antimicrobials used in poultry production among BPF (*n* = 908).

**Figure 3 animals-13-03135-f003:**
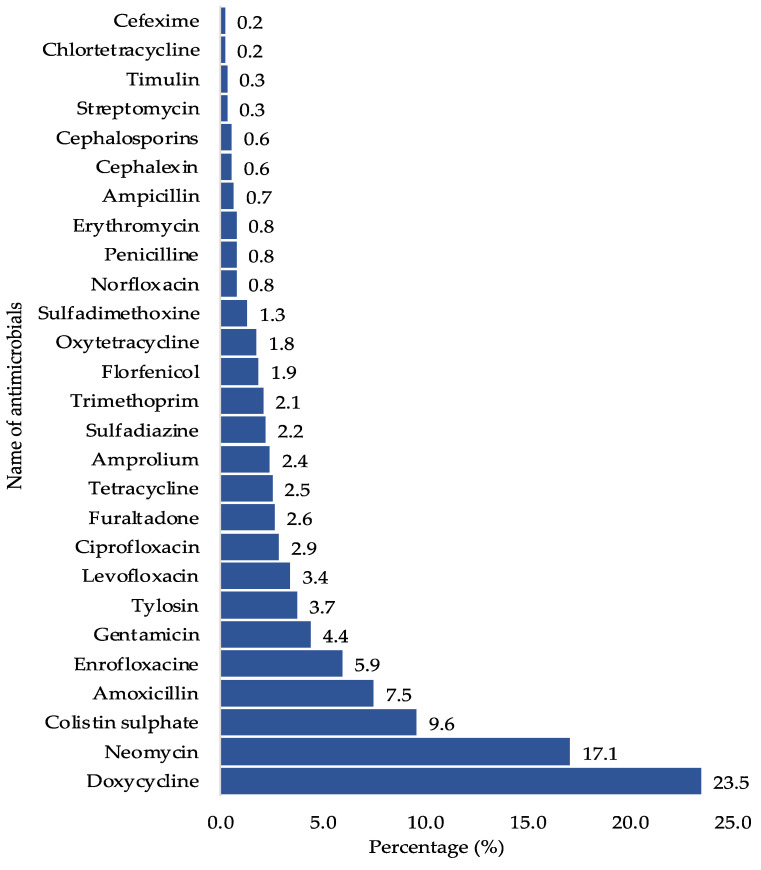
Frequency of AMU in BPF of Nepal (*n* = 908).

**Figure 4 animals-13-03135-f004:**
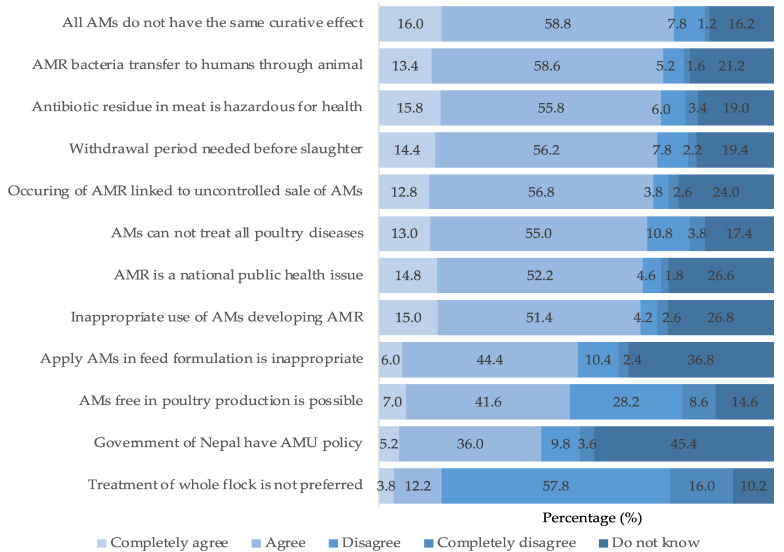
Knowledge of BPF on AMU and AMR. Note: AMs, antimicrobials.

**Figure 5 animals-13-03135-f005:**
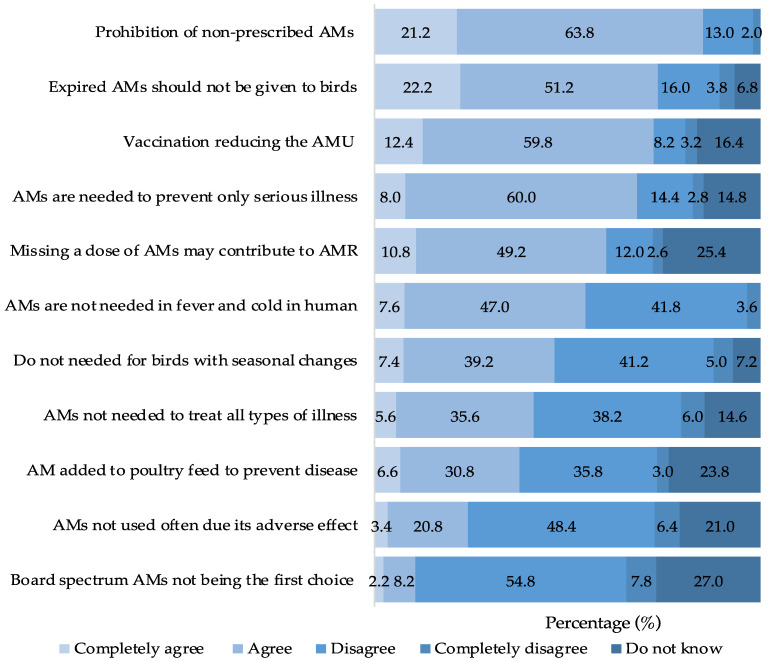
Attitude of BPF toward AMU and AMR.

**Figure 6 animals-13-03135-f006:**
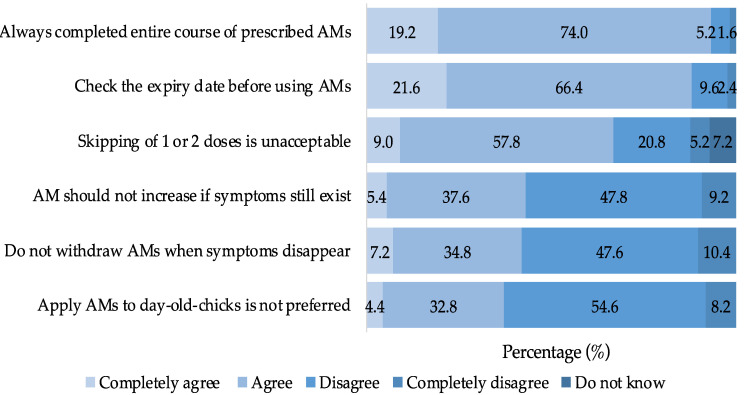
Practice of BPF on AMU and AMR.

**Table 1 animals-13-03135-t001:** Demographic distribution and farm management on the BPF (*n* = 500).

Variable		N (%)
Ecozone	Terai	269 (53.8)
	Hill	211 (42.2)
	Mountain	20 (4.0)
Province	Bagmati	158 (31.6)
	Lumbini	86 (17.2)
	Koshi	85 (17.0)
	Gandaki	63 (12.6)
	Madhesh	51 (10.2)
	Sudurpaschim	30 (6.0)
	Karnali	27 (5.4)
Gender	Male	405 (81.0)
	Female	95 (19.0)
Age (year)	**18–30**	**134 (26.8)**
	31–40	207 (41.4)
	41–60	155 (31.0)
	>60	4 (0.8)
Farming experience (year)	0–4	298 (59.6)
	5–8	135 (27.0)
	9–12	43 (8.6)
	>13	24 (4.8)
Education level of respondent	Illiterate	43 (8.6)
	Primary school	134 (26.8)
	High school	254 (50.8)
	Graduate	69 (13.8)
Flock size (bird head)	Small (<1500)	289 (57.8)
	Medium (≥1500–5000)	147 (29.4)
	Large (>5000)	64 (12.8)
Source of feed	Commercial	493 (98.6)
	Homemade	7 (1.4)
Source of water used for animals	Groundwater	237 (47.4)
	Municipality	206 (41.2)
	Deep well	54 (10.8)
	Other	3 (0.6)

**Table 2 animals-13-03135-t002:** Situation analysis of AMU in BPF (*n* = 500).

Variable		N (%)
Purpose of antimicrobials used in feed (*n* = 682) *	Treatment	496 (72.7)
	Prevention and control	124 (18.2)
	Growth promotion	21 (3.1)
	All above	26 (3.8)
	Others	15 (2.2)
Source of antimicrobials (*n* = 619) *	Drug seller	310 (50.1)
	Veterinarian	207 (33.4)
	Para-veterinarian	64 (10.3)
	Free purchase from a drug seller	18 (2.9)
	Others	20 (3.2)
Frequently observed clinical signs (*n* = 647) *	Digestive system	256 (39.6)
	Respiratory system	244 (37.7)
	Nervous system	22 (3.4)
	Immune system	10 (1.5)
	Skin and integument system	2 (0.3)
	Others	113 (17.5)
Age of bird that used antimicrobial for treatment (day) (*n* = 781) *	0–3	130 (16.6)
	4–10	96 (12.3)
	11–20	198 (25.4)
	21–30	171 (21.9)
	>30	186 (23.8)
Source of consultation when birds are sick (*n* = 500)	Veterinarian	266 (53.2)
	Drug dealer	108 (21.6)
	Para-veterinarian	100 (20.0)
	Self-treatment by farmer	22 (4.4)
	Other	4 (0.8)

Note: * Multiple-choice answers were allowed.

**Table 3 animals-13-03135-t003:** Univariate logistic regression analysis of risk factors associated with KAP of BPF on AMU and AMR.

Variable		Knowledge		Attitude		Practice	
		OR (95% CI)	*p*	OR (95% CI)	*p*	OR (95% CI)	*p*
Province	Bagmati	Ref.		Ref.		Ref.	
	Gandaki	2.6 (0.9–7.6)	0.07	0.4 (0.1–1.8)	0.20	0.9 (0.4–2.4)	0.90
	Karnali	0.4 (0.1–1.9)	0.20	3.1 (0.6–15.7)	0.20	0.3 (0.07–1.2)	0.08
	Lumbini	0.2 (0.07–0.7)	0.008 *	0.2 (0.03–0.8)	0.03 *	0.7 (0.3–1.3)	0.30
	Madhesh	0.3 (0.08–1.2)	0.10	empty		1.2 (0.5–2.8)	0.70
	Koshi	0.2 (0.1–0.7)	0.009 *	1.6 (0.5–4.7)	0.80	2.9 (1.3–6.1)	0.006 *
	Sudurpaschim	0.2 (0.06–0.9)	0.04 *	3.8 (0.6–24.9)	0.20	0.4 (0.1–1.4)	0.20
Age group (year)	18–30	Ref.		Ref.		Ref.	
	31–40	1.1 (0.4–3.2)	0.80	4.2 (1.2–14.7)	0.02 *	0.6 (0.3–1.2)	0.20
	41–60	1.6 (0.5–5.3)	0.40	2.3 (0.6–8.4)	0.10	1.2 (0.5–2.4)	0.70
	>60	0.9 (0.004–1.7)	0.90	2.2 (0.04–101.0)	0.60	0.5 (0.02–9.4)	0.60
Purpose of AMU	All-purpose	Ref.		Ref.		Ref.	
	Prevention and control	2.2 (0.7–6.9)	0.20	5.9 (1.5–27.7)	0.02 *	1.7 (0.7–4.2)	0.20
	Growth promotion	1.1 (0.1–13.9)	0.90	2.1 (0.7–6.2)	0.20	2.9 (0.2–33.6)	0.40
	Treatment	0.8 (0.4–1.9)	0.70	empty		2.1 (1.1–3.9)	0.01 *
	Do not know	16.4 (1.4–189.5)	0.02 *	3.9 (0.8–18.3)	0.08	0.4 (0.1–1.9)	0.20
Sales control of AMs contributes to AMR	Agree	Ref		Ref.		Ref.	
	Disagree	14.5 (6.7–31.3)	<0.0001 **	2.6 (1.1–6.7)	<0.0001 **	1.4 (0.6–3.3)	0.40
Vaccination can reduce AMU	Agree	Ref		Ref.		Ref.	
	Disagree	3.5 (1.6–7.8)	0.002 *	15.0 (5.3–42.5)	<0.0001 **	0.7 (0.3–1.5)	0.30
Source of advice when poultry are sick	Drug seller	Ref		Ref.		Ref.	
	Para-veterinarian	0.8 (0.2–2.9)	0.80	0.6 (0.04–8.9)	0.70	0.4 (0.1–1.1)	0.07
	Self	5.7 (0.9–34.1)	0.05	0.06 (0.001–11.5)	0.30	8.7 (2.3–32.4)	0.001 *
	Veterinarian	1.7 (0.5–5.3)	0.30	0.2 (0.02–2.5)	0.20	21.0 (9.2–47.9)	<0.0001 **
	Other	20.9 (0.04–9543)	0.30	0.8 (0.002–378.9)	0.90	5.7 (0.4–75.0)	0.20

Note: Ref., reference group; *, *p* < 0.05; **, *p* < 0.0001.

**Table 4 animals-13-03135-t004:** Multivariate logistic regression analysis of risk factors associated with KAP of BPF on AMU and AMR.

Variable		Knowledge		Attitude		Practice	
		OR (95% CI)	*p*	OR (95% CI)	*p*	OR (95% CI)	*p*
Province	Bagmati	Ref		Ref.		Ref.	
	Gandaki	2.6 (0.9–7.6)	0.07	0.4 (0.1–1.8)	0.20	0.9 (0. 4–2.4)	0.90
	Karnali	0.4 (0.1–1.9)	0.20	3.1 (0.6–15.7)	0.20	0.3 (0.07–1.2)	0.08
	Lumbini	0.2 (0.07–0.7)	0.008 *	0.2 (0.03–0.8)	0.03 *	0.7 (0.3–1.3)	0.30
	Madhesh	0.3 (0.08–1.2)	0.10	empty		1.2 (0.5–2.8)	0.70
	Koshi	0.2 (0.1–0.7)	0.009 *	1.6 (0.5–4.7)	0.80	2.9 (1.3–6.1)	0.006 *
	Sudurpaschim	0.2 (0.06–0.9)	0.04 *	3.8 (0.6–24.9)	0.20	0.4 (0.1–1.4)	0.20
Age group (years)	18–30	Ref		Ref.		Ref.	
	31–40	1.1 (0.4–3.2)	0.80	4.2 (1.2–14.7)	0.02 *	0.6 (0.3–1.2)	0.20
	41–60	1.6 (0.5–5.3)	0.40	2.3 (0.6–8.4)	0.10	1.2 (0.5–2.4)	0.70
	>60	0.9 (0.004–1.7)	0.90	2.2 (0.04–101.0)	0.60	0.5 (0.02–9.4)	0.60
Purpose of AMU	All-purpose	Ref.		Ref.		Ref.	
	Prevention and control	2.2 (0.7–6.9)	0.20	5.9 (1.5–27.7)	0.02 *	1.7 (0.7–4.2)	0.20
	Growth promotion	1.1 (0.1–13.9)	0.90	2.1 (0.7–6.2)	0.20	2.9 (0.2–33.6)	0.40
	Treatment	0.8 (0.4–1.9)	0.70	empty		2.1 (1.1–3.9)	0.01 *
	Do not know	16.4 (1.4–189.5)	0.02 *	3.9 (0.8–18.3)	0.08	0.4 (0.1–1.9)	0.20
Sales control of AMs contributes to AMR	Agree	Ref		Ref.		Ref.	
	Disagree	14.5 (6.7–31.3)	<0.0001 **	2.6 (1.1–6.7)	<0.0001 **	1.4 (0.6–3.3)	0.40
Vaccination can reduce AMU	Agree	Ref		Ref.		Ref.	
	Disagree	3.5 (1.6–7.8)	0.002 *	15.0 (5.3–42.5)	<0.0001 **	0.7 (0.3–1.5)	0.30
Source of advice when poultry are sick	Drug seller	Ref		Ref.		Ref.	
	Para-veterinarian	0.8 (0.2–2.9)	0.80	0.6 (0.04–8.9)	0.70	0.4 (0.1–1.1)	0.07
	Self	5.7 (0.9–34.1)	0.05	0.06 (0.001–11.5)	0.30	8.7 (2.3–32.4)	0.001 *
	Veterinarian	1.7 (0.5–5.3)	0.30	0.2 (0.02–2.5)	0.20	21.0 (9.2–47.9)	<0.0001 **
	Other	20.9 (0.04–9543)	0.30	0.8 (0.002–378.9)	0.90	5.7 (0.4–75.0)	0.20

Note: Ref., reference group; *, *p* < 0.05; **, *p* < 0.0001.

**Table 5 animals-13-03135-t005:** Association among the KAP on AMU and AMR of farmers.

Variable	Adjusted OR (95% C.I.)	*p*
Knowledge and attitude	19.4 (6.7–55.6)	<0.0001 **
Knowledge and practice	1.7 (1.2–2.5)	0.004 *
Attitude and practice	1.1 (0.6–2.1)	0.070

Note: *, *p* < 0.05; **, *p* < 0.0001.

## Data Availability

The data used in this study are available within the manuscript.

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
