# Peer review of "Situational Analysis and Knowledge, Attitudes, and Practices of Antimicrobial Use and Resistance among Broiler Poultry Farmers in Nepal"

_animals, 2023, doi:10.3390/ani13193135_

Round 1

Reviewer 1 Report

The manuscript “Situation Analysis on Antimicrobial Use and Resistance among Broiler Poultry Farmers of Nepal: Knowledge, Attitudes, and Practices” evaluated the knowledge, attitudes, and practices (KAP) of broiler poultry farmers (BPF) in Nepal concerning the use of antimicrobials (AMU) and antimicrobial resistance (AMR).

In recent time use of antimicrobials in livestock is a global concern due to emergence of antimicrobial resistance (AMR). Low- and middle-income countries are facing greater AMR challenges compared to other developed countries. Thus, increasing awareness of antimicrobial usage (AMU) among various stakeholders is extremely important. In that context, situation analysis of the existing KAP regarding AMU and AMR in animal health is very much warranted.  Therefore, the current study is very much relevant.

 Introduction section nicely introduced the subject and narrated the major objective of the study.

 A total of 500 BPF from 40 districts participated in the study, therefore sample size is good. In materials and methods, sufficient details have been provided.

 Results have been nicely presented and discussed.

 Overall, the work is very important, rigorous and well presented. The writing is also very good. I do not have any concern with the work.    

Minor editing is required. 

Author Response

Thank you for reviewing our manuscript. We have made revisions based on the reviewer's feedback and have submitted the manuscript to Enago for English language editing. Additionally, we have included point-by-point responses to address the revisions.

Reviewer 1

- In recent time use of antimicrobials in livestock is a global concern due to emergence of antimicrobial resistance (AMR). Low- and middle-income countries are facing greater AMR challenges compared to other developed countries. Thus, increasing awareness of antimicrobial usage (AMU) among various stakeholders is extremely important. In that context, situation analysis of the existing KAP regarding AMU and AMR in animal health is very much warranted.  Therefore, the current study is very much relevant. Introduction section nicely introduced the subject and narrated the major objective of the study. A total of 500 BPF from 40 districts participated in the study, therefore sample size is good. In materials and methods, sufficient details have been provided. Results have been nicely presented and discussed. Overall, the work is very important, rigorous and well presented. The writing is also very good. I do not have any concern with the work.   

Thank you for your comments.

- Quality of English Language

              Enago is responsible for the review and certification of the English language.

Reviewer 2 Report

In this study, authors conducted KAP analysis on AMR and AMU and surveyed 500 poultry farmers from 40 districts in all seven provinces of Nepal. The knowledge and practices were limited but the attitude towards AMU and AMR were positive. I am interesting to learn that the most used antimicrobials on farm included doxycycline, neomycin and colistin sulfate. I would like to see more discussions on regulation of antibiotics in food animals in Nepal and compare with other LMICs in the region. Overall, article contains interesting scientific information and some references are needed to be updated, see my comments below. I have few comments

Line 27: Please the rephrase as “Data regarding antimicrobial use and resistance were collected from a

total of 500 BPF from 40 districts, accounting for 27 for 88.1% of Nepal’s broiler poultry population”.

Line 57: Author may cite a study from Sagor et al., http://www.pvj.com.pk/pdf-files/42_2/153-160.pdf

Line 62: Author may support this statement by citing Malik et al., 2022 (http://www.pvj.com.pk/pdf-files/42_3/383-389.pdf) and Zulqarnain et al. http://www.pvj.com.pk/pdf-files/41_2/284-288.pdf

Line 411-412: Most used antimicrobials on farm included doxycycline (23.5%), neomycin (17.1%), and colistin sulfate (9.6%) which are similar to other LMICs and should be compared and  discussed with citations https://www.mdpi.com/2079-6382/10/5/598 and https://www.tandfonline.com/doi/full/10.1080/16549716.2019.1697541.

Line 439-444: Should discuss with recent results from Pakistan, Bangladesh and Nigeria by Umair et al., Lancet Microbes)  

Author Response

Thank you for reviewing our manuscript. We have made revisions based on the reviewer's feedback and have submitted the manuscript to Enago for English language editing. Additionally, we have included point-by-point responses to address the revision.

Reviewer 3 Report

Specific comments

1.      Please write in more detail about the purpose of this study.

2.      Please explain the process of setting each item of the survey.

3.      Do you have any expected results for each analysis item before the survey? To what extent do the predicted values match the survey results?

4.      Line 78. Even if abbreviations are defined in the abstract, all abbreviations must be defined within the text. Please provide a definition of the abbreviation for “KAP” in the text.

5.      Recheck your manuscript for typos or grammatical errors.

Author Response

(The authors gave the same response as above.)

Reviewer 4 Report

The article is well-written and designed. Nowadays all studies on the wide use of antimicrobials are essential and crucial.

In this study, the number of respondents was very satisfactory, so the given conclusions are justified. The results are well described and presented on readable graphs and tables. The literature is well selected and most of the items come from the last few years.

Only a few minor comments have been listed below:

line 59 - Sweden is the European country

line 60-62 - this sentence sounds like praise and an advantage of using antibiotics as growth stimulants, there is no indicated negative impact on the environment, animal health, human health, and the possibility of diagnosing bacterial diseases

line 245 - why farmers used antimicrobials as growth promotion, as in Nepal a growth promotion was banned from 2017?

line 313 - I'm not supposed that antimicrobials are consumed? of course oral administration is mainly used, but it is not consumption

Discussion section - in the part, where the authors described the use of antimicrobials in poultry production, the impact of the wide use of antimicrobials on public health was not underlined

Author Response

(The authors gave the same response as above.)

Reviewer 5 Report

It is an interesting study on the situation analysis on antimicrobial use, but I disagree with the title because there is actually no data on antibiotic resistance.

In fact, in the summary of the article the conclusion indicates …”this study was to conduct a situation analysis of the existing KAP 101 concerning AMU and AMR in animal health. The results of this study have potential to 102 provide guidance to policymakers to combat AMR, nationally and global”……

The introduction of the article is adequate and the material and methods section is very well described.

The results are in accordance with the objectives and are well summarized in the tables.

Does not provide data on antimicrobial resistance, but it is not the objective of the study

The discussion is extensive and well documented.

The conclusions are too broad and aspects already mentioned in the discussion section are repeated. They must be more specific when writing them. I suggest modifying 

Author Response

(The authors gave the same response as above.)
